# CXCL9, CXCL10, and CXCL11; biomarkers of pulmonary inflammation associated with autoimmunity in patients with collagen vascular diseases–associated interstitial lung disease and interstitial pneumonia with autoimmune features

**Masami Kameda[1‡], Mitsuo Otsuka[2‡], Hirofumi Chiba[1‡], Koji Kuronuma[1]\*, Takehiro Hasegawa[3], Hiroki Takahashi[4], Hiroki Takahashi[1]**

1 Department of Respiratory Medicine and Allergology, Sapporo Medical University School of Medicine, Sapporo, Hokkaido, Japan, 2 Department of Respiratory Medicine, Sapporo-Kosei General Hospital, Sapporo, Hokkaido, Japan, 3 Sysmex Corporation, Kobe, Japan, 4 Department of Rheumatology, Sapporo Medical University School of Medicine, Sapporo, Hokkaido, Japan

‡ These authors share first authorship on this work.
\* kuronumak@sapmed.ac.jp

## Abstract

### Introduction

Interstitial lung disease (ILD) is a heterogeneous group of diseases characterized by varying degrees of lung inflammation and/or fibrosis. We investigated biomarkers to infer whether patients with collagen vascular diseases associated ILD (CVD–ILD) and interstitial pneumonia with autoimmune features (IPAF) benefit from immunosuppressive therapy.

### Materials and methods

We retrospectively investigated patients with CVD–ILD, IPAF, and idiopathic pulmonary fibrosis (IPF) between June 2013 and May 2017 at our department. First, we assessed differences in serum and bronchoalveolar lavage fluid (BALF) levels of cytokines between groups. Second, we assessed the associations of patient's clinical variables with serum and BALF levels of those cytokines that were different between groups. Finally, we assessed the associations of diagnosis and response to immunosuppressive therapy with serum levels of those cytokines that were different between groups.

### Results

We included 102 patients (51 with IPF, 35 with IPAF, and 16 with CVD–ILD). Serum and BALF levels of CXCL9, CXCL10, and CXCL11 were significantly elevated in patients with IPAF or CVD–ILD compared with those in patients with IPF. BALF levels of CXCL9 and CXCL10 were correlated with the percentages of lymphocytes and macrophages in BALF. Serum levels of CXCL9 and CXCL10 were correlated with BALF levels. Serum levels of

**Data Availability Statement:** All relevant data are within the manuscript and its Supporting Information files.

**Funding:** MK, MO, and HT received funding from Sysmex Corp. TH is an employee of Sysmex Corp. T.H. contributed to the measurement of cytokines. The funder provided support in the form of salaries for TH, but did not have any additional role in the study design, data collection and analysis, decision to publish, or preparation of the manuscript.

**Competing interests:** Masami Kameda, Mitsuo Otsuka, and Takehiro Hasegawa have a pending patent application belonging to Sysmex Corporation and Sapporo Medical University. This does not alter our adherence to PLOS ONE policies on materials.

CXCL9, CXCL10, and CXCL11 were correlated C-reactive protein, percent predicted forced vital capacity, alveolar-arterial oxygen difference, and the percentages of lymphocytes and macrophages in BALF. Serum levels of CXCL9, CXCL10, and CXCL11 showed moderate accuracy to distinguish patients with CVD–ILD from those with IPAF and IPF. Pre-treatment serum levels of CXCL9 and CXCL11 showed strong positive correlations with the annual forced vital capacity changes in patients with IPAF and CVD–ILD treated with immunosuppressive drugs.

## Conclusions

Serum CXCL9, CXCL10, and CXCL11 are potential biomarkers for autoimmune inflammation and predictors of the immunosuppressive therapy responses in ILD with background autoimmunity.

## Introduction

Interstitial lung disease (ILD) is a heterogeneous group of lung diseases characterized by a combination of inflammation and fibrosis of the diffuse lung parenchyma. The clinical course and immunosuppressive therapeutic response vary substantially among the different ILD types. Idiopathic pulmonary fibrosis (IPF) is the most common ILD characterized by chronic progressive fibrosis and a poor prognosis [1]. In contrast, collagen vascular diseases (CVDs) are heterogeneous diseases characterized by systemic autoimmunity and varying degrees of inflammation and immune-mediated organ damage. Patients with CVD associated ILD (CVD–ILD) have a more favorable clinical course than those with IPF. Therefore, to distinguish IPF and CVD–ILD is important in clinical practice [1]. However, some patients with ILD have features of CVD, that do not meet the classification criteria for a defined CVD. Consequently, the European Respiratory Society/American Thoracic Society (ERS/ATS) proposed a new classification, interstitial pneumonia with autoimmune features (IPAF), to provide an initial framework for evaluating these patients and detecting those who may benefit from immunosuppressive therapy [2]. In clinical practice, the differential diagnosis of ILDs with background autoimmunity and other ILDs remains difficult, because it requires multidisciplinary discussion. Most CVDs can respond to immunosuppressive therapy, suggesting inflammation is a pathological mechanism of this disease. Cytokines are important for CVD pathogenesis, especially C-X-C motif chemokine (CXCL)9, CXCL10, and CXCL11 create local amplification loops responsible for sustaining inflammation in target organs [3]. These three chemokines are ligands for CXCR3 and recruit CD4+ Th1 cells and CD8+ T cells to the site of tissue damage and inhibit angiogenesis during pulmonary fibrosis [4]. Studies have shown that these three chemokines play roles in the inflammatory pathophysiology of CVD and ILD [3, 5–7]. However, the roles of autoimmunity and inflammation in the pathogenesis of CVD–ILD and IPAF remain unclear, and consensus guidelines for their diagnosis or treatment are not available. Therefore, biomarkers that can evaluate autoimmune inflammation and be useful for ILD diagnosis and management are needed [8–10].

We conducted a verifiable analysis based on serum and bronchoalveolar lavage fluid (BALF) CXCL9, CXCL10, CXCL11, and other cytokines in patients with CVD–ILD, IPAF, and IPF. We compared cytokines levels among patients with CVD–ILD, IPAF, and IPF. We then assessed the associations of CXCL9, CXCL10, and CXCL11, that were different between groups, with clinical characteristics, diagnosis, and treatment responsiveness to identify biomarkers of inflammation in ILD with background autoimmunity.

## Materials and methods

### Study design and characteristics of participants

We performed a single center retrospective cohort study of 102 patients with CVD–ILD, IPAF, or IPF at the Department of Respiratory Medicine and Allergology, Sapporo Medical University Hospital between June 2013 and May 2017. We reviewed the records of patients for the diagnosis of ILD and diagnostic criteria for CVD, IPAF, and IPF. We only included untreated patients at diagnosis to avoid influence on cytokines levels. ILD was diagnosed by a pulmonologist with chest high-resolution computed tomography (HRCT). CVD was diagnosed by a rheumatologist according to American College of Rheumatology/European League Against Rheumatism classification criteria [11–15], Alarcon-Segovia Diagnostic Criteria for Mixed Connective Tissue Disease [16], or EMEA vasculitis classification algorithm [17], and CVD–ILD was diagnosed by two pulmonologists. IPAF was diagnosed by two pulmonologists according to the ERS/ATS classification criteria [2]. IPF was diagnosed by two pulmonologists according to the ATS/ERS/Japanese Respiratory Society/Latin American Thoracic Association classification criteria [18]. We excluded patients didn't undergo initial examinations, those with known causes other than CVD (e.g. infection, asbestosis, hypersensitivity pneumonitis, drug-related pneumonitis), those with severe other disease, and those with nonIPF or nonIPAF. Fig 1 shows patient selection flowchart. The patients suspected to have ILD underwent initial examinations, physical and laboratory examinations, arterial blood gas analyses (BGA), pulmonary functional tests (PFTs), high-resolution computed tomographies (HRCTs), and BALFs within three months of diagnosis. All the patients had signed informed consents at diagnosis to allow the collection of serum and BALF samples for future studies, and serum and BALF samples were obtained at initial examinations and stored at −80˚C until use. The patients who received ILD treatments after diagnosis underwent physical and laboratory

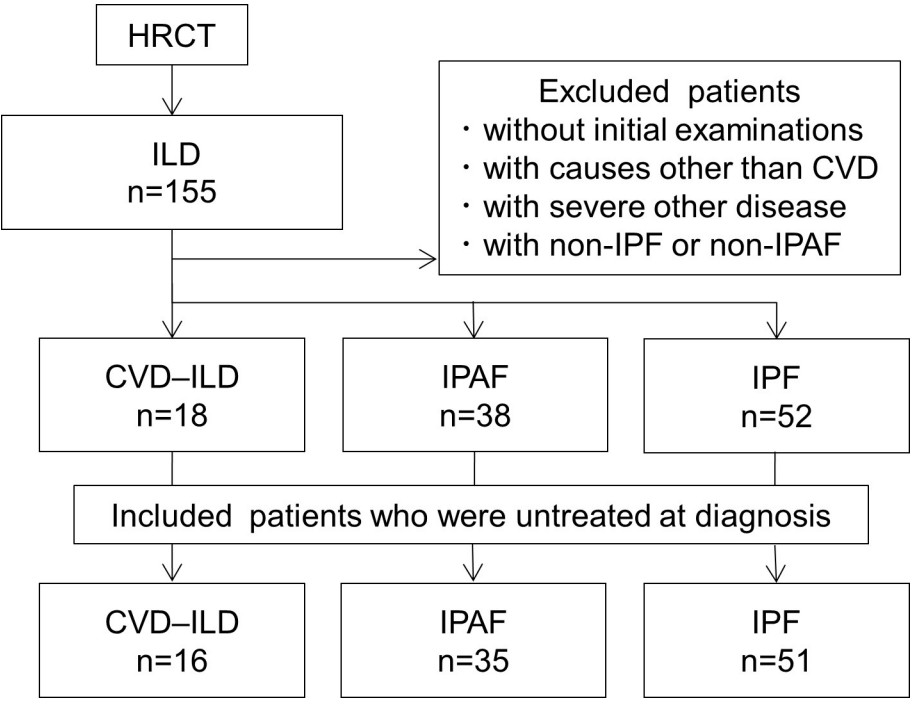

**Fig 1. Patient selection flowchart.**

examination, BGA, PFTs, and HRCTs 12 ± 3 months after the treatment. The Institutional Review Board of Sapporo Medical University Hospital approved this study (No. 282–160, approved on 12/13/2016).

## Aim

The aim of our study was to identify biomarkers of inflammation in ILD with background autoimmunity for the diagnosis and treatment.

## BAL processing

A flexible bronchoscope, wedged into a subsegmental bronchus of the right middle lobe or the left lingula, was used to infuse 50 ml of 0.9% saline at body temperature and to collect a lavage specimen by applying gentle suction. This process was repeated three times, and the BAL specimens were pooled together.

## Biomarker analysis

We curated putative cytokines from the literature as potentially being involved in the pathogenesis of CVD–ILD, IPAF, and IPF. Serum and BALF samples were analyzed for cytokines levels, including CC chemokine ligand (CCL)3, CCL7, CCL17, CXCL9, CXCL10, CXCL11, fas-ligand (Fas-L), interferon (IFN)γ, interleukin (IL)-18, IL-4, IL-6, IL-8, IL-10, IL-17, tumor necrosis factor (TNF)α, and tumor necrosis factor superfamily member (TNFSF)14. CCL3, CCL17 and CXCL9 were measured in the automated immunoassay system HISCL® (Sysmex, Kobe, Japan), and other cytokines were measured with chemiluminescence ELISAs. CCL7, CXCL10, CXCL11, Fas-L, IFNγ, IL-18, IL-4, IL-6, IL-8, IL-10, IL-17, TNFα, and TNFSF14 were measured using a sandwich ELISA system with the following antibodies: anti-IL-4 (8D4-8 and MP4-25D2) and anti-IFNγ (NIB42 and 4S.B3) purchased from BioLegend (CA, USA), anti-IL-17A (eBio64CAP17 and eBio64DEC17) and anti-IL-10 (JES3-9D7 and JES3-12G8) purchased from eBioscience (CA, USA), and anti-IL-18 (159-12B and 125-2H) purchased from MBL (Aichi, Japan). CCL7, CXCL10, CXCL11, Fas-L, IL-6, IL-8, TNFα, and TNFSF14 were measured using an ELISA development system DuoSet (R&D Systems). Recombinant cytokines, including IL-4 (BD Biosciences, NJ, USA), IL-10, IL-17A, IFNγ (BioLegend), and IL-18 (MBL) were used as standards. All detection protocols were modified by using streptavidin–alkaline phosphatase (Vector Laboratories, CA, USA) and CDP-Star Substrate with Sapphire-II Enhancer (Life Technologies). The chemiluminescence intensity was measured on an Infinite® 200 PRO microplate reader (Tecan Group Ltd, Männedorf, Switzerland).

## Statistical analysis

To verify biomarkers reflect inflammation in ILD with background autoimmunity, we assessed differences in characteristics and cytokines levels between the CVD–ILD, IPAF, and IPF groups using the Kruskal–Wallis test with the Steel–Dwass post-hoc test, or Fisher's exact test with the post-hoc Benjamini–Hochberg test. We expressed continuous variables as medians with interquartile ranges and categorical variables as numbers or percentages. To assess the associations of CXCL9, CXCL10, and CXCL11 with the clinical characteristics of ILD, we calculated the correlations between serum and bronchoalveolar lavage fluid (BALF) levels of these chemokines and patient's clinical variables using the Spearman's rank correlation coefficient. To evaluate the value of serum CXCL9, CXCL10 and CXCL11 as diagnostic biomarkers for ILD with background autoimmunity, we constructed receiver operating characteristic (ROC) curves. To evaluate the value of serum CXCL9, CXCL10 and CXCL11 as predictive

biomarkers of treatment responsiveness of ILD with background autoimmunity, we calculated correlations between pre-treatment serum levels of these chemokines and the annual forced vital capacity (FVC) changes using the Spearman's rank correlation coefficient. We performed the Benjamini–Hochberg test using the R software (R Development Core Team, Vienna, Austria) and other analyses using the JMP version 10.0 software (SAS Institute, Cary, NC, USA). We set statistical significance at $p < 0.05$.

## Results

### Comparison of patients' clinical characteristics

Among the 102 patients, we identified 16 with CVD–ILD, 35 with IPAF, and 51 with IPF. The CVDs included systemic sclerosis (SSc) (n = 5), rheumatoid arthritis (RA) (n = 4), polymyositis (PM) (n = 2), microscopic polyangiitis (n = 2), mixed connective tissue disease (n = 2), Sjögren's syndrome (n = 1)). Table 1 shows a comparison of patients' clinical characteristics between the CVD–ILD, IPAF, and IPF groups. We found no significant differences in terms of

**Table 1. Comparison of baseline clinical characteristics between the CVD–ILD, IPAF, and IPF groups.**

| Clinical characteristics | CTD–ILD | IPAF | IPF | p-value | | |
|---|---|---|---|---|---|---|
| | | | | CTD–ILD vs IPAF | CTD–ILD vs IPF | IPAF vs IPF |
| Subject, n | 16 | 35 | 51 | | | |
| Age, years | 71.1 (61.7–74.0) | 72.8 (62.8–76.8) | 68.6 (63.3–73.2) | 0.882 | 0.939 | 0.687 |
| Females, n (%) | 11 (68.8) | 20 (57.1) | 12 (23.5) | 0.543 | 0.004* | 0.004* |
| Smoker, n (%) | 11 (68.8) | 23 (65.7) | 41 (80.4) | 1.000 | 0.490 | 0.419 |
| Onset to diagnosis, months | 6 (4–15) | 5 (3–15) | 8.0 (3–48) | 0.886 | 0.768 | 0.470 |
| HRCT pattern | | | | | | |
| UIP, n (%) | 7 (43.7) | 5 (14.3) | 51 (100.0) | | | |
| NSIP, n (%) | 5 (31.3) | 16 (45.7) | 0 (0.0) | | | |
| Other/unclassifiable, n(%) | 4 (25.0) | 14 (40.0) | 0 (0.0) | | | |
| Pulmonary function test | | | | | | |
| FVC, %predicted | 80.8 (71.0–101.3) | 82.2 (71.4–92.1) | 100.9 (79.4–123.7) | 0.947 | 0.144 | 0.002* |
| DLco, %predicted | 54.9 (39.2–62.0) | 53.9 (45.3–61.4) | 61.2 (46.0–76.7) | 0.740 | 0.166 | 0.134 |
| A–aDO$_2$, torr | 22.6 (12.9–33.0) | 15.5 (10.3–23.6) | 12.8 (8.1–16.8) | 0.266 | 0.013* | 0.264 |
| BALF, n | 14 | 29 | 43 | | | |
| Retrieved rate, % | 48.0 (33.5–57.0) | 62.7 (46.7–72.0) | 57.3 (44.0–64.0) | 0.024* | 0.112 | 0.142 |
| Cell concentration, ×10$^5$/ml | 1.2 (0.6–2.1) | 1.2(0.7–2.1) | 1.1 (0.6–1.9) | 0.996 | 0.867 | 0.843 |
| Macrophages, % | 78.0 (70.9–86.5) | 76.3 (64.6–84.1) | 85.8 (80.2–91.6) | 0.944 | 0.053 | 0.007* |
| Lymphocytes, % | 7.8 (4.8–17.4) | 15.0 (8.2–22.4) | 9.0 (3.6–12.6) | 0.328 | 0.994 | 0.029* |
| Neutrophils, % | 6.6 (4.0–8.5) | 2.9 (1.7–8.1) | 2.2 (1.0–3.7) | 0.404 | 0.009* | 0.203 |
| Eosinophils, % | 2.6 (0.9–4.5) | 1.6 (0.5–5.8) | 1.6 (0.3–3.9) | 0.944 | 0.563 | 0.714 |
| CD4/CD8 ratio | 0.9 (0.7–1.4) | 1.5 (0.8–2.7) | 2.0 (1.2–3.0) | 0.255 | 0.012* | 0.262 |
| Laboratory data, n | 16 | 35 | 51 | | | |
| CRP, mg/dL | 0.29 (0.0–1.04) | 0.12 (0.0–0.51) | 0.0 (0.0––0.28) | 0.258 | 0.047* | 0.561 |
| SP-A, ng/ml | 68.9 (59.5–102.4) | 64.0 (47.8–92.6) | 55.8 (41.8–102.8) | 0.498 | 0.287 | 0.709 |
| SP-D, ng/ml | 207(141–292) | 228 (170–354) | 207 (120–281) | 0.606 | 0.951 | 0.226 |
| KL-6, U/ml | 948 (831–1449) | 1120 (608–1926) | 810 (400–1340) | 0.997 | 0.164 | 0.0496* |

Data are presented as counts (n) or medians and ranges (interquartile range). HRCT: high-resolution computed tomography; UIP: usual interstitial pneumonia; NSIP: nonspecific interstitial pneumonia; FVC: forced vital capacity; DLco: diffusing capacity of the lung for carbon monoxide; A–aDO$_2$: alveolar-arterial oxygen difference; CRP: C-reactive protein; SP: surfactant protein; KL: krebs von den lungen. BALF: bronchoalveolar lavage fluid. *$p < 0.05$.

clinical characteristics between the CVD–ILD and IPAF groups. The proportions of women were significantly higher in the CVD–ILD and IPAF groups (68.8% and 57.1%) than in the IPF group (23.5%). The predicted percentages of forced vital capacity (%FVC) were significantly lower in the IPAF group (82.2%) than in the IPF group (100.9%). The alveolar–arterial oxygen differences (A–$aDO_2$) were significantly higher in the CVD–ILD group (22.6 torr) than in the IPF group (12.8 torr). In the BALF examination, the percentages of lymphocytes (lymphocytes %) were significantly higher and the percentages of macrophages (macrophages %) were lower in the IPAF group (15.0% and 76.3%) than in the IPF group (9.0% and 85.8%). The percentages of neutrophils (neutrophils %) were significantly higher and the CD4/CD8 ratios were significantly lower in the CVD–ILD group (6.6% and 0.9) than in the IPF group (2.2% and 2.0). C-reactive protein (CRP) levels were significantly higher in the CVD–ILD and IPAF groups (0.29 and 0,12 mg/dL) than in the IPF group (0.0 mg/dL). Krebs von den Lungen (KL)-6 levels were significantly higher in the IPAF group (1120 U/ml) than in the IPF group (810 U/ml). %FVCs, macrophage%, and BALF CD4/CD8 ratios tended to be lower, and A–$aDO_2$, BALF neutrophils%, and CRP levels tended to be higher in the order of IPF, IPAF, and CVD–ILD group. S1 Table shows organ involvement in CVD–ILD group. In all patients of the CVD–ILD group, the most severely damaged organ was the lung.

## Serum cytokines level comparisons

Table 2 shows a comparison of the serum cytokines levels between the CVD–ILD, IPAF, and IPF groups. We found significant differences in terms of serum levels of CXCL9, CXCL10, CXCL11, IL-6, IL-10, and TNFα between the CVD–ILD, IPAF, and IPF groups (CXCL9; 82.4, 38.0, and 28.7 pg/ml, CXCL10; 381.2, 141.3, and 80.4 pg/ml, CXCL11; 182.7, 52.6, and 0.0 pg/ml, IL-6 1.5, 0.5, and 0.3 pg/ml, IL-10; 17.4, 14.3, and 9.1 pg/ml, TNFα; 20.2, 11.0, and 6.7 pg/ml). Serum levels of CXCL9, CXCL10, and CXCL11 were higher in order of the CVD–ILD, IPAF, and IPF groups. Serum levels of IFNγ, IL-4, and IL-17 were below the detection limit in more than 75% of the cases.

**Table 2. Comparison of baseline serum cytokines levels between the CVD–ILD, IPAF, and IPF groups.**

| | CVD–ILD | IPAF | IPF | p-value | | |
|---|---|---|---|---|---|---|
| | | | | CVD–ILD vs IPAF | CVD–ILD vs IPF | IPAF vs IPF |
| CCL3, pg/ml | 19.3 (14.3–26.6) | 15.1 (11.4–23.9) | 13.4 (11.2–19.8) | 0.233 | 0.036* | 0.892 |
| CCL7, pg/ml | 8.8 (6.0–14.7) | 8.5 (4.6–15.1) | 6.5 (3.6–13.2) | 0.923 | 0.378 | 0.626 |
| CCL17, pg/ml | 555.2 (318.1–699.8) | 462.9 (311.0–752.2) | 555.0 (418.3–810.9) | 0.988 | 0.653 | 0.274 |
| CXCL9, pg/ml | 82.4 (34.2–174.3) | 38.0 (21.1–88.6) | 28.7 (20.1–56.0) | 0.042* | 0.002* | 0.342 |
| CXCL10, pg/ml | 381.2 (163.1–991.5) | 141.3 (51.2–258.5) | 80.4 (23.1–150.7) | 0.014* | <0.001* | 0.022* |
| CXCL11, pg/ml | 182.7 (91.4–348.5) | 52.6 (0.0–117.2) | 0.0 (0.0–25.3) | 0.008* | <0.001* | 0.007* |
| Fas-L, pg/ml | 65.5 (52.1–79.6) | 61.0 (49.0–70.6) | 56.8 (44.6–67.3) | 0.801 | 0.159 | 0.347 |
| IL-6, pg/ml | 1.5 (0.5–3.2) | 0.5 (0.0–1.3) | 0.3 (0.0–1.3) | 0.045* | 0.011* | 0.952 |
| IL-8, pg/ml | 22.7 (12.4–32.7) | 18.0 (13.4–23.2) | 19.7 (13.7–33.7) | 0.897 | 0.996 | 0.875 |
| IL-10, pg/ml | 17.4 (12.9–23.8) | 14.3 (9.7–19.7) | 9.1 (6.0–13.2) | 0.192 | <0.001* | 0.016* |
| IL-18, pg/ml | 658.6 (355.0–1159.5) | 423.2 (313.4–581.9) | 398.4 (318.4–487.3) | 0.300 | 0.061 | 0.606 |
| TNFα, pg/ml | 20.2 (16.7–41.8) | 11.0 (0.3–26.2) | 6.7 (0.0–15.3) | 0.268 | 0.004* | 0.177 |
| TNFSF14, pg/ml | 50.6 (20.9–132.8) | 43.6 (13.7–66.8) | 47.0 (10.7–102.7) | 0.946 | 0.917 | 0.989 |

Data are presented as medians and ranges (interquartile range). CCL: CC chemokine ligand; CXCL: C-X-C motif chemokine; Fas-L: fas-ligand; IL: interleukin; TNF: tumor necrosis factor; TNFSF14: tumor necrosis factor superfamily member. *$p < 0.05$.

## BALF cytokines level comparisons

Table 3 shows a comparison of BALF cytokines levels between the CVD–ILD, IPAF, and IPF groups. We found significant differences in terms of BALF levels of CXCL9, CXCL10, CCL3, Fas-L, IL-6, IL-8, and IL-18 between the CVD–ILD, IPAF, and IPF groups (CXCL9; 11.1, 5.8, and 3.0 pg/ml, CXCL10; 99.0, 102.8, and 37.8 pg/ml, CCL3; 8.0, 5.2, and 3.7 pg/ml, Fas-L; 0.7, 1.6, and 0.5 pg/ml, IL-6; 0.5, 0.5, and 0.1 pg/ml, IL-8; 157.4, 97.0, and 48.5 pg/ml, IL-18; 12.4, 12.3, and 4.5 pg/ml). However, the BALF cytokines levels between the CVD–ILD and IPAF groups were similar. BALF levels of CXCL9 and CXCL10 were higher in the IPAF and CVD–ILD groups. BALF levels of IFNγ, IL-4, and IL17 were below the detection limit in more than 75% of the cases.

## Associations between serum and BALF cytokines levels

S2 Table shows the associations between serum and BALF cytokines levels. Between serum and BALF levels, CXCL9 showed a moderate correlation (rs = 0.43) and CXCL10 showed a weak correlation (rs = 0.39).

## Associations between CXCL9, CXCL10, and CXCL11 levels and clinical characteristics

Table 4a shows the association between serum CXCL9, CXCL10, and CXCL11 levels and clinical characteristics. Serum levels of CXCL10 and CXCL11 showed weak negative correlations with %FVC (rs = −0.31 and −0.38) and weak positive correlations with A–aDO$_2$ (rs = 0.23 and 0.40). Serum levels of CXCL9, CXCL10, and CXCL11 showed weak positive correlations with CRP levels (rs = 0.36, 0.36, and 0.37). Serum levels of CXCL9 and CXCL10 showed weak negative correlations and CXCL11 showed a moderate negative correlation with BALF macrophages% (rs = −0.38, −0.33, and −0.49). Serum levels of CXCL9, CXCL10, and CXCL11 showed weak positive correlations with BALF lymphocytes% (rs = 0.32, 0.23, and 0.37). Serum levels of CXCL9 and CXCL10 showed weak negative correlations with BALF CD4/CD8 ratios (rs = −0.26 and −0.26).

**Table 3. Comparison of baseline BALF cytokines levels between the CVD–ILD, IPAF, and IPF groups.**

| | CVD–ILD | IPAF | IPF | p-value | | |
|---|---|---|---|---|---|---|
| | | | | CVD–ILD vs IPAF | CVD–ILD vs IPF | IPAF vs IPF |
| CCL3, pg/ml | 8.0 (4.7–13.4) | 5.2 (2.5–20.2) | 3.7 (2.1–6.1) | 0.962 | 0.027* | 0.128 |
| CCL7, pg/ml | 2.6 (1.9–4.3) | 2.7 (1.5–4.0) | 2.0 (1.2–3.7) | 0.983 | 0.437 | 0.489 |
| CCL17, pg/ml | 4.3 (2.0–7.0) | 5.0 (3.0–8.7) | 5.6 (3.1–9.9) | 0.715 | 0.361 | 0.817 |
| CXCL9, pg/ml | 11.1 (3.0–14.1) | 5.8 (3.5–21.3) | 3.0 (1.0–7.0) | 0.998 | 0.057 | 0.013* |
| CXCL10, pg/ml | 99.0 (55.5–180.3) | 102.8 (58.4–181.6) | 37.8(9.8–84.6) | 0.999 | 0.028* | 0.007* |
| CXCL11, pg/ml | 3.3 (0.0–9.3) | 1.5 (0.0–5.8) | 3.5 (0.0–7.6) | 0.936 | 0.996 | 0.733 |
| Fas-L, pg/ml | 0.7 (0.3–2.7) | 1.6 (0.7–3.2) | 0.5 (0.0–1.5) | 0.361 | 0.323 | 0.005* |
| IL-6, pg/ml | 0.5 (0.3–3.7) | 0.5 (0.2–5.0) | 0.1 (0.0–0.4) | 0.991 | 0.015* | 0.001* |
| IL-8, pg/ml | 157.4 (65.6–266.4) | 97.0 (37.4–269.9) | 48.5 (28.5–110.6) | 0.676 | 0.015* | 0.138 |
| IL-10, pg/ml | 0.0 (0.0–2.6) | 0.0 (0.0–5.2) | 0.0 (0.0–0.2) | 0.995 | 0.695 | 0.652 |
| IL-18, pg/ml | 12.4 (5.3–28.6) | 12.3 (3.2–33.1) | 4.5 (2.4–8.9) | 0.959 | 0.067 | 0.128 |
| TNFα, pg/ml | 3.1 (0.0–22.2) | 0.0 (0.0–17.1) | 4.9 (0.0–12.7) | 0.893 | 0.946 | 0.970 |
| TNFSF14, pg/ml | 0.0 (0.0–2.5) | 0.0 (0.0–13.7) | 0.0 (0.0–22.2) | 0.582 | 0.767 | 0.985 |

Data are presented as or medians and ranges (interquartile range). BALF: bronchoalveolar lavage fluid. CCL: CC chemokine ligand; CXCL: C-X-C motif chemokine; Fas-L: fas-ligand; IL: interleukin; TNF: tumor necrosis factor; TNFSF14: tumor necrosis factor superfamily member. *p < 0.05.

**Table 4. Association between baseline CXCL9, CXCL10, and CXCL11 levels and clinical characteristics.**

a. Serum CXCL9, CXCL10, and CXCL11 levels

| | rs | | | | | | | | |
|---|---|---|---|---|---|---|---|---|---|
| | %FVC | %DLco | A–aDO$_2$ | CRP | BALF Macrophages% | BALF Lymphocytes% | BALF Neutrophils% | BALF Eosinophils% | BALF CD4/8 ratio |
| CXCL9 | −0.17 | −0.11 | 0.23* | 0.36* | −0.38* | 0.32* | 0.10 | 0.22* | −0.26* |
| CXCL10 | −0.31* | −0.20 | 0.40* | 0.36* | −0.33* | 0.23* | 0.19 | 0.26* | −0.26* |
| CXCL11 | −0.38* | −0.20 | 0.35* | 0.37* | −0.49* | 0.37* | 0.33* | 0.16 | −0.21 |

b. BALF CXCL9, CXCL10, and CXCL11 levels

| | rs | | | | | | | | |
|---|---|---|---|---|---|---|---|---|---|
| | %FVC | %DLco | A–aDO$_2$ | CRP | BALF Macrophages% | BALF Lymphocytes% | BALF Neutrophils% | BALF Eosinophils% | BALF CD4/8 ratio |
| CXCL9 | −0.28* | −0.11 | 0.28* | 0.19 | −0.67* | 0.53* | 0.37* | 0.42* | 0.08 |
| CXCL10 | −0.27* | −0.17 | 0.24* | 0.19 | −0.60* | 0.51* | 0.29* | 0.32* | 0.10 |
| CXCL11 | 0.05 | 0.19 | 0.10 | −0.08 | −0.11 | 0.07 | 0.17 | −0.02 | 0.20 |

CXCL: C-X-C motif chemokine; %FVC: percent predicted forced vital capacity; %DLco: percent predicted diffusing capacity of the lung for carbon monoxide; A–aDO$_2$: alveolar-arterial oxygen difference; CRP: C-reactive protein; BALF: bronchoalveolar lavage fluid. *$p < 0.05$.

Table 4b shows associations between BALF CXCL9, CXCL10, and CXCL11 levels and their clinical characteristics. BALF levels of CXCL9 and CXCL10 showed weak negative correlations with %FVC (rs = −0.28 and −0.27) and weak positive correlations with A–aDO$_2$ (rs = 0.28 and 0.24). BALF levels of CXCL9 and CXCL10 showed moderate negative correlations with BALF macrophages% (rs = −0.67 and −0.60) and moderate positive correlations with BALF lymphocytes% (rs = 0.53 and 0.51), and weak positive correlations BALF neutrophils% (rs = 0.37 and 0.29), and BALF eosinophils% (rs = 0.42 and 0.32).

## Associations between serum CXCL9, CXCL10, and CXCL11 levels and diagnoses in the CVD–ILD, IPAF, and IPF groups

Table 5 shows the results of the ROC curve to assess the diagnostic value of serum CXCL9, CXCL10, and CXCL11 for the differential diagnosis of the CVD–ILD, IPAF, and IPF groups.

**Table 5. Associations between serum cytokine levels and diagnoses in the CVD–ILD, IPAF, and IPF groups.**

| | Sensitivity | Specificity | Cut-off value | AUC |
|---|---|---|---|---|
| CVD–ILD vs IPAF | | | | |
| CXCL9, pg/ml | 0.87 | 0.49 | 32.5 | 0.72 |
| CXCL10, pg/ml | 0.60 | 0.80 | 287.1 | 0.75 |
| CXCL11, pg/ml | 0.87 | 0.69 | 89.9 | 0.77 |
| CVD–ILD vs IPF | | | | |
| CXCL9, pg/ml | 0.67 | 0.87 | 71.9 | 0.79 |
| CXCL10, pg/ml | 0.73 | 0.90 | 203.2 | 0.89 |
| CXCL11, pg/ml | 0.87 | 0.94 | 89.9 | 0.90 |
| IPAF vs IPF | | | | |
| CXCL9, pg/ml | 1.00 | 0.04 | 208.8 | 0.41 |
| CXCL10, pg/ml | 0.40 | 0.96 | 226.6 | 0.67 |
| CXCL11, pg/ml | 0.54 | 0.81 | 48.3 | 0.68 |

CXCL: C-X-C motif chemokine.

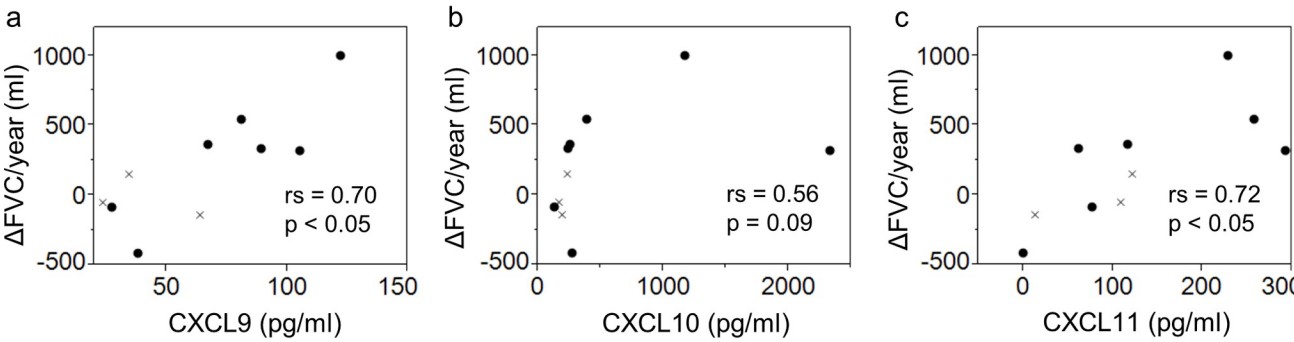

**Fig 2. Associations between serum cytokines and treatment responsiveness in the CVD–ILD and IPAF groups.** Associations between pretreatment serum CXCL9, CXCL10, and CXCL11 levels and annual FVC changes in the CVD–ILD and IPAF groups. a) CXCL9, b) CXCL10, and c) CXC11. The p-values were estimated using Spearman's rank correlation coefficient. CXCL: C-X-C motif chemokine; FVC: forced vital capacity; ×: CVD–ILD; •: IPAF.

To distinguish CVD–ILD from IPAF and IPF groups, serum CXCL9, CXCL10, and CXCL11 showed moderate accuracy with AUC ranged 0.72–0.90. To distinguish IPAF from IPF groups, serum CXCL10 and CXCL11 levels showed low accuracy with AUC ranged 0.67–0.68.

## Associations between serum CXCL9, CXCL10, and CXCL11 and treatment responsiveness in the CVD–ILD and IPAF groups

Fig 2 shows the associations between pre-treatment serum cytokines levels and annual FVC changes among 10 patients who had been treated with corticosteroid and/or immunosuppressive drugs. Pre-treatment serum levels of CXCL9 and CXCL11 showed strong positive correlations with the annual FVC changes after treatment (rs = 0.70 and 0.72).

## Discussion

This study demonstrated that in patients with CVD–ILD and IPAF with background autoimmunity, serum and BALF levels of CXCL9, CXCL10, and CXCL11 were significantly higher than those in patients with IPF. The serum levels of CXCL9 and CXCL10 were significantly associated with the BALF levels. The serum levels of CXCL9, CXCL10, and CXCL11 were correlated with increases in BALF lymphocytes%, CRP levels, and A–aDO$_2$. In addition, BALF levels of CXCL9 and CXCL10 were correlated with increases in A–aDO$_2$ and BALF lymphocytes%. Serum CXCL9, CXCL10, and CXCL11 showed moderate accuracy to distinguish CVD–ILD from IPAF and IPF. Pre-treatment serum levels of CXCL9 and CXCL11 showed strong positive correlations with the annual FVC changes among patients treated with immunosuppressive drugs. These findings suggest that CXCL9, CXCL10, and CXCL11 may be biomarkers of autoimmune inflammation in patients with CVD–ILD and IPAF.

We showed that the serum levels of CXCL, CXCL10, and CXCL11 and the BALF levels of CXCL9 and CXCL10 were higher in patients with CVD–ILD and IPAF than in those with IPF. CXCL9, CXCL10, and CXCL11 are involved in the pathogenesis of CVD–ILD and ILD as follows: In patients with PM/DM and SSc, the serum CXCL10 levels are higher in patients with ILD than in those without ILD [19, 20]. In the case of PM/DM–ILD, patients with anti-Jo-1 antibody showed higher serum CXCL9 and CXCL10 levels than patients with IPF [21]. In terms of RA–ILD, serum CXCL9, CXCL10, and CXCL11 are increased and induce CXCR3+ T cells in the lung [22, 23]. In ILD, CXCL9, CXCL10, and CXCL11 levels are involved in inflammation through the induction of CXCR3+ T cells [5, 6]. In IPAF, diagnosis requires patients

presenting ILD, not meeting the CVD classification criteria, and having at least one feature from 2 of the 3 domains (clinical, serologic, and morphologic) [2]. IPAF has been shown to manifest with clinical extrathoracic autoimmune features in 47.3% to 62.5% of patients [8]. These results suggest that CXCR3+ T cells related inflammation is activated in the lungs and systemically in patients with CVD–ILD and IPAF, respectively.

This study showed that in patients with CVD–ILD, IPAF, and IPF, the BALF levels of CXCL9 and CXCL10 were correlated with increases in A–aDO$_2$ and BALF lymphocytes% and that serum levels of CXCL9, CXCL10, and CXCL11 were correlated with increases in BALF lymphocytes%, CRP levels, and A–aDO$_2$. We consider that these three chemokines may induce pulmonary inflammation through lymphocyte induction that lead to alveolitis resulting diffusion impairment in patients with CVD–ILD and IPAF. In particular, the serum CXCL9 and CXCL10 levels in our patients showed correlations with BALF levels so that these serum chemokines may reflect not only systemic inflammation but also local pulmonary inflammation. On the other hand, BALF CXCL11 levels, unlike CXCL9 and CXCL10, were similar among CVD–ILD, IPAF, and IPF patients and had no correlations with BALF or serum levels. This may be due to the insufficient measurement sensitivity for CXCL11 because the BALF levels of CXCL11 were below the detection limit in many patients. As the levels of CXCL9, CXCL10, and CXCL11 differed between the CVD–ILD, IPAF, and IPF groups, we expected the correlations between these chemokines and characteristics may differ between groups. However, because of retrospective study, we were unable to adjust pulmonary function and retrieved rates of BALF between the groups and could not assess differences in the correlations caused by background autoimmunity in a subgroup analysis (S3 Table).

Our AUCs for serum CXCL9, CXCL10 and CXCL11 in the differential diagnosis of CVD–ILD from IPAF and IPF suggest that these serum chemokines may be diagnostic biomarkers of ILD with autoimmunity. In addition, we showed that patients with CVD–ILD and IPAF treated with immunosuppressants had higher pre-treatment serum levels of CXCL9 and CXCL11 and greater improvements in annual FVC after treatment. Studies have also suggested associations between the disease activity of autoimmune diseases and serum or BALF CXCL9, CXCL10, and CXCL11 levels in patients with SLE, DM, or SScs [7, 20, 24, 25]. However, the clinical management of CVD–ILD and IPAF remains difficult due to a lack of accurate diagnosis, disease activity, and therapeutic response markers [2, 8]. Our results suggest that high serum CXCL9, CXCL10, and CXCL11 levels may reflect reversible inflammation and that these chemokines are predictive biomarkers of the response to immunosuppressive therapy in ILD with background autoimmunity. Especially, serum CXCL9 may be a novel independent biomarker because it does not correlate with clinical ILD biomarkers, surfactant protein (SP)-A, SP-D, or KL-6 (S4 Table) [26]. In the future, monitoring of these serum biomarkers from the time of diagnosis could be useful in determining whether immunosuppressive therapy is initiated without BAL. In particular, when ILD worsens during follow-up, these biomarkers may enable clinicians to choose anti-inflammatory drugs or anti-fibrotic drugs.

We found that the cytokines profile of IPAF was similar to that of CVD–ILD. However, we found a difference in terms of the serum levels, especially serum CXCL9, CXCL10, and CXCL11 levels in the IPAF group were intermediate between those in the IPF and the CVD–ILD group. Autoimmune inflammation in CVD–ILD occurs systemically. In comparison, in IPAF, pulmonary inflammation and some extrathoracic autoimmune features that are not enough for a CVD diagnosis occurs [8, 10]. Thus, serum CXCL9, CXCL10, and CXCL11 levels may reflect the activities of autoimmune inflammation and the existence of autoimmune features.

We are aware of the limitations of our study. First, we studied only a small number of patients at a single center. The number of CVD–ILD patients was relatively small, and CVD

included different diseases. Second, many patients in the IPF group had mild diseases with small decrease in FVCs. Japanese clinicians actively perform HRCTs in patients suspected to have ILDs, because anti-fibrotic therapies for IPF are effective during the early stages of the disease. Consequently, our cohort included many early stage IPF patients. Furthermore, these 3 chemokines showed no differences depending on the severity of IPF (S5 Table). Third, we evaluated only a few treated patients. Twenty-seven of the 41 patients in the CVD–ILD and IPAF groups were treated with immunosuppressive drug, but only 10 of them had pre-treatment serum samples and respiratory function test data one year after treatment due to the retrospective study nature. Further studies are needed to evaluate the utility of CXCL9, CXCL10, and CXCL11 as biomarkers of treatment responsiveness. Fourth, we could not confirm the presence of CXCR3+ T cells in lung tissues, so our results are based on indirect proof. Ideal biomarkers should be easily and repeatedly collected.

## Conclusions

Serum CXCL9, CXCL10, and CXCL11 may reflect autoimmune inflammation of ILD and work as biomarkers to predict the response to immunosuppressive therapy in the management of ILD with background autoimmunity.

## Supporting information

**S1 Table. Organ involvement in CVD–ILD.** CK: Creatine kinase; SSc: systemic sclerosis; RA: rheumatoid arthritis; PM: polymyositis; MPA: microscopic polyangiitis; MCTD: mixed connective tissue disease; SjS: Sjögren's syndrome.
(DOCX)

**S2 Table. Associations between serum and BALF biomarker levels.** CCL: CC chemokine ligand; CXCL: C-X-C motif chemokine; Fas-L: fas-ligand; IL: inter-leukin; TNF: tumor necrosis factor; TNFSF14: tumor ne-crosis factor superfamily member. $^*p < 0.05$.
(DOCX)

**S3 Table. Associations between baseline CXCL9, CXCL10, and CXCL11 levels and clinical characteristics in the CVD–ILD, IPAF, and IPF groups.** CXCL: C-X-C motif chemokine; % FVC: percent predicted forced vital capacity; %DLco: percent predicted diffusing capacity of the lung for carbon monoxide; A–aDO2: alveolar-arterial oxygen difference; CRP: C-reactive protein; BALF: bronchoalveolar lavage fluid. $^*p < 0.05$.
(DOCX)

**S4 Table. Associations between serum CXCL9, CXCL10, and CXCL11 levels and main IPF biomarkers levels.** CXCL: C-X-C motif chemokine;SP: surfactant protein; KL: krebs von den lungen. $^*p < 0.05$.
(DOCX)

**S5 Table. Associations between serum CXCL9, CXCL10, and CXCL11 levels and the severity of IPF.** CXCL: C-X-C motif chemokine. $^*p < 0.05$.
(DOCX)

## Acknowledgments

The authors gratefully acknowledge Sysmex Corporation for measuring cytokines levels. We also acknowledge Maho Yoshida, Takami Kondo, Takeshi Aritsu, and Yasuhiro Otomo of Sysmex Corporation for their support of our activity, Kunihide Ino (President, ACCPREC Inc.,

Takatsuki, Japan) for support of statistical analysis, and Enago (www.enago.jp) for the English language review.

## Author Contributions

**Conceptualization:** Masami Kameda, Mitsuo Otsuka, Hiroki Takahashi.

**Data curation:** Masami Kameda, Mitsuo Otsuka, Hirofumi Chiba.

**Formal analysis:** Masami Kameda, Mitsuo Otsuka, Hirofumi Chiba.

**Investigation:** Hirofumi Chiba, Takehiro Hasegawa, Hiroki Takahashi.

**Methodology:** Takehiro Hasegawa.

**Supervision:** Koji Kuronuma, Hiroki Takahashi, Hiroki Takahashi.

**Writing – original draft:** Masami Kameda, Mitsuo Otsuka, Koji Kuronuma.

**Writing – review & editing:** Mitsuo Otsuka, Koji Kuronuma.

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
