## [Decision Letter · Decision Letter 0]

8 Sep 2020

PONE-D-20-22949

CXCL9, CXCL10, and CXCL11; Biomarkers of pulmonary inflammation associated with autoimmunity in patients with collagen vascular diseases–associated interstitial lung disease and interstitial pneumonia with autoimmune features.

PLOS ONE

Dear Dr. Kuronuma,

Thank you for submitting your manuscript to PLOS ONE. After careful consideration, we feel that it has merit but does not fully meet PLOS ONE’s publication criteria as it currently stands. Therefore, we invite you to submit a revised version of the manuscript that addresses the points raised during the review process.

Our reviewers found some interests in this study, but also pointed out a number of criticisms that are useful for improving this manuscript. I ask the authors to fully respond to all comments made by reviewers in the revised version.

We look forward to receiving your revised manuscript.

Kind regards,

Masataka Kuwana, MD, PhD

Academic Editor

PLOS ONE

Journal Requirements:

2. Please consider modifying the title to ensure that it is meeting PLOS’ guidelines (https://journals.plos.org/plosone/s/submission-guidelines#loc-title). In particular, the title should be specific, descriptive and, particularly in this case, concise.

4.Thank you for providing the following Funding Statement: 

[MK, MO, and HT received funding from Sysmex Corp. TH is an employee of Sysmex Corp.

T.H. contributed to the measurement of cytokines.

The funder had no control over the interpretation, writing, or publication of this work.].

We note that one or more of the authors is affiliated with the funding organization, indicating the funder may have had some role in the design, data collection, analysis or preparation of your manuscript for publication; in other words, the funder played an indirect role through the participation of the co-authors.

If the funding organization did not play a role in the study design, data collection and analysis, decision to publish, or preparation of the manuscript and only provided financial support in the form of authors' salaries and/or research materials, please review your statements relating to the author contributions, and ensure you have specifically and accurately indicated the role(s) that these authors had in your study in the Author Contributions section of the online submission form. Please make any necessary amendments directly within this section of the online submission form.  Please also update your Funding Statement to include the following statement: “The funder provided support in the form of salaries for authors [insert relevant initials], but did not have any additional role in the study design, data collection and analysis, decision to publish, or preparation of the manuscript. The specific roles of these authors are articulated in the ‘author contributions’ section.”

If the funding organization did have an additional role, please state and explain that role within your Funding Statement.

Please also provide an updated Competing Interests Statement declaring this commercial affiliation along with any other relevant declarations relating to employment, consultancy, patents, products in development, or marketed products, etc.  

We note that you have a patent relating to material pertinent to this article. Please provide an amended statement of Competing Interests to declare this patent (with details including name and number), along with any other relevant declarations relating to employment, consultancy, patents, products in development or modified products etc. Please confirm that this does not alter your adherence to all PLOS ONE policies on sharing data and materials, as detailed online in our guide for authors http://journals.plos.org/plosone/s/competing-interests by including the following statement: "This does not alter our adherence to  PLOS ONE policies on sharing data and materials.” If there are restrictions on sharing of data and/or materials, please state these. Please note that we cannot proceed with consideration of your article until this information has been declared.

Reviewers' comments:

Reviewer's Responses to Questions

**Comments to the Author**

1. Is the manuscript technically sound, and do the data support the conclusions?

Reviewer #1: Yes

Reviewer #2: Partly

2. Has the statistical analysis been performed appropriately and rigorously? 

Reviewer #1: Yes

Reviewer #2: Yes

3. Have the authors made all data underlying the findings in their manuscript fully available?

Reviewer #1: Yes

Reviewer #2: Yes

4. Is the manuscript presented in an intelligible fashion and written in standard English?

Reviewer #1: Yes

Reviewer #2: Yes

5. Review Comments to the Author

Reviewer #1: The authors evaluated the cytokine levels in serum and BALF, and found that serum CXCL9, CXCL10, and CXCL11 are useful biomarkers for autoimmune inflammation and predictors of immunosuppressant responses in ILD. Although the effect of the anti-fibrotic drug, nintedanib, on progressive fibrosing ILD has been already established, anti-inflammatory drugs are also effective in some ILD, including CVD-ILD and IPAF. However, there is currently no established biomarkers and treatments for them (anti-inflammatory drug or anti-fibrotic drug, or both?). This study provides an important contribution to the biomarker for patients who use anti-inflammatory drugs.

<major comments="">

The authors described that CXCL9, CXCL10, and CXCL11 are important for the diagnosis of CVD-ILD and IPAF and anti-inflammatory therapy. However, it is unclear how to use these biomarkers in clinical practice. It would be informative to discuss the clinical application of these biomarkers.

<minor comments="">

L141 cytokiness→cytokines

P11L182 Indicate the actual measurement value in the results (same below)</minor></major>

Reviewer #2: <general comments="">

In this manuscript, Kameda and colleagues assessed serum and BALF levels of various cytokines, including CXCL9, CXCL10, and CXCL11, in patients with CVD-ILD, IPAF and IPF. They also assessed clinical significance of these cytokine levels (CXCL9, CXCL10, and CXCL11) in serum and BALF, regarding association of clinical variables, diagnoses and response to immunosuppressive therapy. They found that serum and BALF levels of CXCL9, CXCL10, and CXCL11 were elevated in patients with CVD-ILD or IPAF compared with IPF. Levels of these chemokine were correlated with several clinical variables. In 10 patients with CVD-ILD or IPAF who treated with corticosteroid and/or immunosuppressive agents, serum CXCL9 and CXCL11 levels were positively associated with annual FVC changes. According to these findings, they concluded that CXCL9, CXCL10 and CXCL11 may be biomarkers to evaluate autoimmune inflammation and to predict response to immunosuppressive treatments. Their findings highlight interesting aspects of distinct pathophysiology among CVD-ILD, IPAF and IFP, however, there are several concerns in this study. As described in the limitation by the authors, the number of the patients, especially patients with CVD-ILD, were very small. Additionally, CVD-ILD contain variety of diseases, such as RA, Ssc, PM, MCTD and SS, which have different pathophysiology, respectively. Moreover, the number of patients who treated by immunosuppressive agent were only 10, which was too small to indicate that serum CXCL9 and CXCL11 are potential biomarker to predict response to immunosuppressive therapy.

<comments>

The authors showed the association between CXCL9, CXCL10 and CXCL11 levels and clinical variables, such as %FVC, %DLCO, A-aDO2, CRP and BALF findings in all the patient in Table 4 (page 15, 16). As the levels of these markers differed among CVD-ILD, IPAF and IPF, the correlations between these markers and clinical variables may be different among the three groups. The author should explain these points or assess the correlations in each group, respectively.

The authors showed the association between serum CXCL9, CXCL10 or CXCL11 levels and FVC change in 10 patients who treated with corticosteroid and/or immunosuppressive agents in CVD-ILD and IPAF groups in Figure 2 (page 17). This study include 16 patients with CDV-ILD and 35 patients with IPAF. In that case, how the 41 patients in CVD-ILD and IPAF groups were treated? The majority of the patients (41 patients) in CVD-ILD and IPAF groups did not receive corticosteroid and/or immunosuppressive agents? It is unclear how the authors chose the 10 patients for the correlation analyses between serum markers and response to treatment?</comments></general>

6. PLOS authors have the option to publish the peer review history of their article (what does this mean?). If published, this will include your full peer review and any attached files.

Reviewer #1: No

Reviewer #2: No

---

## [Author Response · Author response to Decision Letter 0]

28 Sep 2020

Reviewer #1: The authors evaluated the cytokine levels in serum and BALF, and found that serum CXCL9, CXCL10, and CXCL11 are useful biomarkers for autoimmune inflammation and predictors of immunosuppressant responses in ILD. Although the effect of the anti-fibrotic drug, nintedanib, on progressive fibrosing ILD has been already established, anti-inflammatory drugs are also effective in some ILD, including CVD-ILD and IPAF. However, there is currently no established biomarkers and treatments for them (anti-inflammatory drug or anti-fibrotic drug, or both?). This study provides an important contribution to the biomarker for patients who use anti-inflammatory drugs.

-> We thank the reviewer 1 for the positive overall evaluation of our study.

The authors described that CXCL9, CXCL10, and CXCL11 are important for the diagnosis of CVD-ILD and IPAF and anti-inflammatory therapy. However, it is unclear how to use these biomarkers in clinical practice. It would be informative to discuss the clinical application of these biomarkers.

-> We appreciate the precise comments. We added the sentence below at lane 361-363.

These serum biomarkers may enable clinicians to more easily evaluate when and to whom to administer anti-inflammatory drugs with simple blood tests in ILD management.

L141 cytokiness→cytokines

-> We corrected.

P11L182 Indicate the actual measurement value in the results (same below)

-> We added the measured value in the results. Please confirm the lanes 178-190, 210-213, 228-231, 246-247, 253-267, and 296.

Reviewer #2: 

In this manuscript, Kameda and colleagues assessed serum and BALF levels of various cytokines, including CXCL9, CXCL10, and CXCL11, in patients with CVD-ILD, IPAF and IPF. They also assessed clinical significance of these cytokine levels (CXCL9, CXCL10, and CXCL11) in serum and BALF, regarding association of clinical variables, diagnoses and response to immunosuppressive therapy. They found that serum and BALF levels of CXCL9, CXCL10, and CXCL11 were elevated in patients with CVD-ILD or IPAF compared with IPF. Levels of these chemokine were correlated with several clinical variables. In 10 patients with CVD-ILD or IPAF who treated with corticosteroid and/or immunosuppressive agents, serum CXCL9 and CXCL11 levels were positively associated with annual FVC changes. According to these findings, they concluded that CXCL9, CXCL10 and CXCL11 may be biomarkers to evaluate autoimmune inflammation and to predict response to immunosuppressive treatments. Their findings highlight interesting aspects of distinct pathophysiology among CVD-ILD, IPAF and IFP, however, there are several concerns in this study. As described in the limitation by the authors, the number of the patients, especially patients with CVD-ILD, were very small. Additionally, CVD-ILD contain variety of diseases, such as RA, Ssc, PM, MCTD and SS, which have different pathophysiology, respectively. Moreover, the number of patients who treated by immunosuppressive agent were only 10, which was too small to indicate that serum CXCL9 and CXCL11 are potential biomarker to predict response to immunosuppressive therapy.

-> We thank the reviewer 2 for the comments and suggestions in our study. 

The authors showed the association between CXCL9, CXCL10 and CXCL11 levels and clinical variables, such as %FVC, %DLCO, A-aDO2, CRP and BALF findings in all the patient in Table 4 (page 15, 16). As the levels of these markers differed among CVD-ILD, IPAF and IPF, the correlations between these markers and clinical variables may be different among the three groups. The author should explain these points or assess the correlations in each group, respectively.

-> We appreciate the comments and suggestions about the correlations between the markers and clinical variables. We also expected the correlations between these chemokines and clinical variables may be different between the CVD–ILD, IPAF, and IPF groups. However, we could not adjust pulmonary function and retrieved rate of BALF between groups because of retrospective study. The normal BALF retrieved rate is 40-70%, but the proportion of patients with BALF retrieved rate of 40% or less in the CVD-ILD group was higher than in the other groups. This made it difficult to evaluate the correlations in the CVD-ILD group. We could not assess differences in the correlations caused by background autoimmunity in a subgroup analysis (S3 Table). We need a subgroup analysis in the next prospective study. We added retrieved rate of BALF in Table1 and the sentence below at lane 343-349.

As the levels of CXCL9, CXCL10, and CXCL11 differed between the CVD–ILD, IPAF, and IPF groups, we expected the correlations between these chemokines and characteristics may differ between groups. However, because of retrospective study, we were unable to adjust pulmonary function and retrieved rates of BALF between the groups and could not assess differences in the correlations caused by background autoimmunity in a subgroup analysis (S3 Table).

The authors showed the association between serum CXCL9, CXCL10 or CXCL11 levels and FVC change in 10 patients who treated with corticosteroid and/or immunosuppressive agents in CVD-ILD and IPAF groups in Figure 2 (page 17). This study include 16 patients with CDV-ILD and 35 patients with IPAF. In that case, how the 41 patients in CVD-ILD and IPAF groups were treated? The majority of the patients (41 patients) in CVD-ILD and IPAF groups did not receive corticosteroid and/or immunosuppressive agents? It is unclear how the authors chose the 10 patients for the correlation analyses between serum markers and response to treatment?

-> We appreciate the precise comments. In this retrospective observational analysis, the number of samples were relatively low, especially in the immunosuppressive treatment group. We need to evaluate the role as the predictive biomarkers in the next prospective study. We added the sentence below at lane 381-384.

Twenty-seven of the 41 patients in the CVD–ILD and IPAF groups were treated with immunosuppressive drug, but only 10 of them had pre-treatment serum samples and respiratory function test data one year after treatment due to the retrospective study nature.

---

## [Decision Letter · Decision Letter 1]

15 Oct 2020

PONE-D-20-22949R1

CXCL9, CXCL10, and CXCL11; Biomarkers of pulmonary inflammation associated with autoimmunity in patients with collagen vascular diseases–associated interstitial lung disease and interstitial pneumonia with autoimmune features.

PLOS ONE

Dear Dr. Kuronuma,

Thank you for submitting your manuscript to PLOS ONE. After careful consideration, we feel that it has merit but does not fully meet PLOS ONE’s publication criteria as it currently stands. Therefore, we invite you to submit a revised version of the manuscript that addresses the points raised during the review process.

This manuscript has been much improved by revisions, but our reviewers suggest some improvement of the Discussion section. I ask the authors to modify the section in the re-revised version.

We look forward to receiving your revised manuscript.

Kind regards,

Masataka Kuwana, MD, PhD

Academic Editor

PLOS ONE

Reviewers' comments:

Reviewer's Responses to Questions

**Comments to the Author**

1. If the authors have adequately addressed your comments raised in a previous round of review and you feel that this manuscript is now acceptable for publication, you may indicate that here to bypass the “Comments to the Author” section, enter your conflict of interest statement in the “Confidential to Editor” section, and submit your "Accept" recommendation.

Reviewer #1: All comments have been addressed

Reviewer #2: All comments have been addressed

2. Is the manuscript technically sound, and do the data support the conclusions?

Reviewer #1: Yes

Reviewer #2: Yes

3. Has the statistical analysis been performed appropriately and rigorously? 

Reviewer #1: Yes

Reviewer #2: Yes

4. Have the authors made all data underlying the findings in their manuscript fully available?

Reviewer #1: Yes

Reviewer #2: Yes

5. Is the manuscript presented in an intelligible fashion and written in standard English?

Reviewer #1: Yes

Reviewer #2: Yes

6. Review Comments to the Author

Reviewer #1: I think the points that I pointed out last time have been improved.

One last comment, do you think that in the future you will not have to measure BALF if you measure CXCL9, CXCL10, and CXCL11 only with serum? From the results of this manuscript, I understood that. Add a vision of what to do when using this marker clinically in the future. I advise you to make the conclusion easier to understood.

Reviewer #2: The authors have responded to the reviewer’s questions adequately. The manuscript has been improved.

Hope to validate their findings in future prospective studies.

7. PLOS authors have the option to publish the peer review history of their article (what does this mean?). If published, this will include your full peer review and any attached files.

Reviewer #1: No

Reviewer #2: No

---

## [Author Response · Author response to Decision Letter 1]

19 Oct 2020

Reviewer #1: I think the points that I pointed out last time have been improved.

One last comment, do you think that in the future you will not have to measure BALF if you measure CXCL9, CXCL10, and CXCL11 only with serum? From the results of this manuscript, I understood that. Add a vision of what to do when using this marker clinically in the future. I advise you to make the conclusion easier to understood.

-> We appreciate the additional comments. We tried to present the usefulness of these serum biomarkers for clinicians. We added the sentence below at lane 361-367 (discussion), 48-50(abstract conclusions) and 392-394 (conclusions).

In the future, monitoring of these serum biomarkers from the time of diagnosis could be useful in determining whether immunosuppressive therapy is initiated without BAL. In particular, when ILD worsens during follow-up, these biomarkers may enable clinicians to choose anti-inflammatory drugs or anti-fibrotic drugs.

Abstract conclusions

Serum CXCL9, CXCL10, and CXCL11 are potential biomarkers for autoimmune inflammation and predictors of the immunosuppressive therapy responses in ILD with background autoimmunity.

Conclusions

Serum CXCL9, CXCL10, and CXCL11 may reflect autoimmune inflammation of ILD and work as biomarkers to predict the response to immunosuppressive therapy in the management of ILD with background autoimmunity.

Reviewer #2: The authors have responded to the reviewer’s questions adequately. The manuscript has been improved.

Hope to validate their findings in future prospective studies.

-> We thank the reviewer 2 for the comments in our study. We plan to validate in the next prospective study.

---

## [Editor Report · Decision Letter 2]

20 Oct 2020

CXCL9, CXCL10, and CXCL11; Biomarkers of pulmonary inflammation associated with autoimmunity in patients with collagen vascular diseases–associated interstitial lung disease and interstitial pneumonia with autoimmune features.

PONE-D-20-22949R2

Dear Dr. Kuronuma,

We’re pleased to inform you that your manuscript has been judged scientifically suitable for publication and will be formally accepted for publication once it meets all outstanding technical requirements.

Kind regards,

Masataka Kuwana, MD, PhD

Academic Editor

PLOS ONE

Additional Editor Comments (optional):

The authors have adequately responded to the comment made by reviewer #1.
---

## [Editor Report · Acceptance letter]

23 Oct 2020

PONE-D-20-22949R2 

CXCL9, CXCL10, and CXCL11; Biomarkers of pulmonary inflammation associated with autoimmunity in patients with collagen vascular diseases–associated interstitial lung disease and interstitial pneumonia with autoimmune features. 

Dear Dr. Kuronuma:

I'm pleased to inform you that your manuscript has been deemed suitable for publication in PLOS ONE. Congratulations! Your manuscript is now with our production department. 

Kind regards, 

on behalf of

Prof. Masataka Kuwana 

Academic Editor

PLOS ONE